# Measurement of Forest Ecological Benefits Based on Big Data

**Hua Li [1], Shuo Liu [2,3,*] and Tijiu Cai [4]**

[1] Department of Biomedicine and Health Sciences, Shanghai Vocational College of Agriculture and Forestry, Shanghai 201699, China; lih@shafc.edu.cn

[2] Heilongjiang Institute of Construction Technology, Harbin 150025, China

[3] College of Wildlife and Protected Area, Northeast Forestry University, Harbin 150040, China

[4] College of Forestry, Northeast Forestry University, Harbin 150040, China; tijiu.cai@nefu.edu.cn

[*] Correspondence: liushuo@hict.edu.cn; Tel.: +86-186-4630-5163

**Abstract:** Socioeconomic development combined with the development and utilization of forest resources have resulted in the forest ecological environment becoming more and more valuable, and its impact on the national economy has also increased. Firstly, based on a modern statistical model, the dependent variable and independent variable set of standardized forest ecological benefits were determined, and the seemingly unrelated forest ecological benefit model was established. According to the alternative market method and market approximation theory, the first and second monetary models of forest ecological benefit value were constructed using the market approximation coefficient, the effective area coefficient, and the physical quantity conversion coefficient. Next, the concept of "whole diffusion" was introduced into the establishment of the forest ecological benefit model, and the whole diffusion model was established to estimate the physical amount of forest ecological benefits. The concepts of classical forest ecological benefit and generalized forest ecological benefit were proposed. Forest management survey data were used to measure forest ecological benefit, and classic ecological benefit and generalized forest benefit were integrated into one system for the first time. The economic value is 94.31 million RMB ¥.

**Keywords:** big data; forest ecological benefit; measurement; National nature reserve





## 1. Introduction

Forest ecosystem services are multifaceted, which is first manifested in the utilization of forest biological resources. Forests are the only important resource that can provide wood, which is closely related to human production and life. Alternatively, forests provide significant ecological services. Forests have obvious diversion and blocking effects on surface runoff, therefore, they can greatly delay the duration of surface runoff, effectively reduce the peak value of runoff, regulate the flood and dry season, and reduce flood and river interruption. The canopy can objectively intercept part of the precipitation, reduce the mechanical damage of raindrops to the soil. The root system can hold the soil and litter and increase the field precipitation, resulting in a better forest microclimate, which has a beneficial impact on the surrounding farmland, grassland, and other ecosystems. Forests protect the wild animals and plants in the system, allowing them to grow well, absorb a large amount of carbon dioxide in the air, release oxygen, purify the atmosphere, conserve water sources, reduce flood and drought disasters, restrain wind and sand, and reduce noise.

Internationally, Costanza et al., in the United States conducted a series of studies on ecosystem services. The most influential is the paper published in Nature [1]. This paper classified and evaluated ecosystem services on a global scale and divided ecosystem services into 17 types. Costanza et al. [2] published another paper on the value of ecosystem services. Holzman [3] studied the service value of ecosystems. Calder [4] found that forest and water guarantee forest benefits exceed water charges, Pereira [5] explored the cost

support of Amazon Forest environmental services, and Rosser [6] studied the value of a forest as a factor of an eco-economic system. Tiemann and Ring [7] developed biophysical indicators for forest ecosystem services. They also studied the challenges and opportunities of aligning forest function mapping and the ecosystem service concept in Germany [8]. Frélichová et al. [9] integrated an assessment of ecosystem services in the Czech Republic. Barrette et al. [10] performed naturalness assessment using forestry maps to validate forest management sustainability. Sorge et al. [11] built a socio-ecological-technical-analytical framework to understand the dynamics of forest ecosystem services governance. Das et al. [12] analyzed the importance-performance of ecosystem services in tribal communities of the Barind region, Eastern India. Maiera et al. [13] had a systematic literature review for securing forest ecosystem service provision in Europe.

To realize sustainable forestry development, except from the economic value of wood, we should assess the ecology-related economic benefits of forests. Frizzle et al. [14] used a spatial Bayesian belief network to quantify trade-offs among forest-related ecosystem services towards achieving sustainable forestry. Dandabathula [15] explored the nexus between Indian forestry and sustainable development goals. Baumgartner [16] studied the complex relationship between sustainable development goals and the forest sector.

In China, Zhou [17] evaluated the ecological value of forests in Heilongjiang Province using the hypothetical market method. Lang et al. [18] performed a study entitled "10 forest ecological benefits measurement theories and methods of forestry ecological engineering". Sun and Lang [19] published an overall estimation of the seemingly unrelated model of generalized forest ecological benefits. Thereafter, Meng and Hou [20] discussed the research progress and diachronic development of forest ecosystem service value accounting theory and evaluation methods. Wang et al. [21] measured and evaluated the forest ecosystem service and its value in China. Zhu et al. [22] summarized and studied estimation methods of forest ecological value. Some Chinese scholars have published articles that estimated the ecological value of China's forests [23]. Xiao et al. [24] made a value assessment of the function of the forest ecosystem services in Chongqing. Feng et al. [25] had a case evaluation of forest ecosystem services in China. Li [26] studied the improvement of forest ecological benefit system construction in Tibet. Shi [27] explored the ecological benefit characteristics and value of returning farmland to a forest project in the middle and upper reaches of the Yangtze River and the Yellow River. Huang [28] analyzed the coupling relationship between multi-functional changes and comprehensive benefits of forest ecosystem in Hulunbeier. Pan [29] discussed the diverse compensation methods of forest ecological benefits in Heilongjiang Province based on the perspective of "government incentive". Liu et al. [30] used the method of "whole diffusion model" to construct the diffusion models of canopy interception, water holding capacity of litter, soil capillary pore water storage, fixing soil, retaining fertilizer, absorbing carbon dioxide, releasing oxygen, restraining wind and sand. Ou et al. [31] expounded on the connotation of trade-offs in forest ecosystem services and clarified the basic characteristics and main steps of trade-off analysis of forest ecosystem services. Then, we sorted the principles, techniques, and main functions of trade-off analysis methods, as well as their scope of application and analyzed the driving roles of natural and human factors in trade-off of the forest ecosystem services. Liu et al. [32] developed a software system for forest ecological benefit evaluation. It was composed of the calculation modules of the forest ecological benefit parameter table, calculation of six classical forest ecological benefit physical quantities, conversion from physical quantity to monetary quantity, calculation of 11 kinds of monetary quantity of generalized forest ecological benefit. Liu et al. [33] constructed a seemingly unrelated model of forest ecological benefit by using stand, meteorological factors, geographical location factors, and other independent variable sets, and taking various ecological benefits of forests as dependent variables. Liu and Shi [34] made an evaluation of water retention services of forest ecosystems in Fujian Province: Comparison between results from the InVEST model and meta-analysis.

Scholars at home and abroad have not yet formed a unified evaluation system, there is a marked deviation in research results, and the values obtained by different scholars vary greatly. However, there is still an opportunity to expand the research on the forest ecological benefits of the National Nature Reserve at home and abroad. This paper aimed to use forest big data for the National Nature Reserve, which spends a lot of human, material, and financial resources to obtain a forest management survey, allowing us to construct a forest seemingly unrelated model based on the modern statistical theory to study the forest ecological benefits of Liangshui National Nature Reserve.

## 2. Methods

### 2.1. Forest Ecological Benefits Seemingly Unrelated Model

Tang et al. [35] assumed that the linear statistical relationship between the random variables $Y_1, \ldots, Yq$ and some factors based on the modern statistical model, that is:

$$Y_j = X_j \beta^{(j)} + e_J \ldots\ldots\ldots\ldots\ldots 1 \leq j \leq q \tag{1}$$

Generally, it can be assumed that the mean value of the error matrix is 0, and its rows are uncorrelated (independent), that is:

$$E(e) = 0 \tag{2}$$

$$\text{cov}(e_{i\bullet}) = \sigma^2 \sum_{q \times q} \tag{3}$$

$$\text{cov}(e_{i\bullet}, e_{j\bullet}) = \underset{q \times q}{0} \ldots\ldots\ldots\ldots 1 \leq i \neq j \leq n \tag{4}$$

The following linear model group can be obtained from Equations (1)–(4):

$$\begin{cases} \underset{n \times 1}{Y_j} = \underset{n \times p_j}{X_j} \underset{pj \times 1}{\beta}^{(j)} + e_{\bullet j} \\ \\ E(e_{\bullet j}) = 0 \qquad \ldots\ldots\ldots 1 \leq i \neq i' \leq n\, 1 \leq j \leq q \\ \text{cov}(e_{i\bullet}) = \sigma^2 \sum_{q \times q} \\ \text{cov}(e_{i\bullet}, e_{i'\bullet}) = \underset{q \times q}{0} \end{cases} \tag{5}$$

where: $e_{i\bullet} = (e_{i1}, \ldots, e_{iq})$.

Equation (5) is called seemingly uncorrelated linear model.

This model was proposed by Tang et al., based on modern statistics. It widened the application scope of the usual univariate or multivariate linear model or nonlinear model. At that time, simultaneous equations were established only for water source cultivation, fixing soil, retaining fertilizer, absorbing carbon dioxide, releasing oxygen, and restraining wind and sand. Based on Tang Shouzheng's seemingly unrelated model, this study will define various classical concepts of forest ecological benefits according to the unique biological characteristics of the forest, adding easily available independent variable sets, such as stand, meteorological factors, and geographical location factors affecting forest growth, and will take various ecological benefits of the forest as dependent variables, to construct a complex seemingly unrelated model of forest ecological benefits.

$$\begin{cases} \ln(y_{i1}) = a_{11} + b_{11}LF_{i1} + b_{21}LF_{i2} + b_{31}LF_{i3} + c_{11}LZ_{i1} + c_{21}LZ_{i2} + c_{31}LZ_{i3} + d_{11} \times ln(JY_i) + d_{21}YB_i + e_{i1} \\ y_{i2} = a_{12} + b_{12}LF_{i1} + b_{22}LF_{i2} + b_{32}LF_{i3} + c_{12}LZ_{i1} + c_{22}LZ_{i2} + c_{32}LZ_{i3} + d_{32}JD_i + d_{42}WD_i + d_{52}HB_i + e_{i2} \\ y_{i3} = a_{13} + b_{13}LF_{i1} + b_{23}LF_{i2} + b_{33}LF_{i3} + d_{33}JD_i + d_{43}WD_i + +d_{53}HB_i + e_{i3} \\ y_{i4} = a_{14} + c_{14}LZ_{i1} + c_{24}LZ_{i2} + c_{34}LZ_{i3} + d_{14}JY_i + d_{24}YB_i + d_{34}JD_i + d_{44}WD_i + e_{i4} \\ y_{i5} = a_{15} + c_{15}LZ_{i1} + c_{25}LZ_{i2} + c_{35}LZ_{i3} + d_{15}JY_i + d_{25}YB_i + d_{35}JD_i + d_{45}WD_i + e_{i5} \\ y_{i6} = a_{16} + c_{16}LZ_{i1} + c_{26}LZ_{i2} + c_{36}LZ_{i3} + d_{16}JY_i + d_{26}YB_i + d_{36}JD_i + d_{46}WD_i + e_{i6} \\ y_{i7} = a_{17} + c_{17}LZ_{i1} + c_{27}LZ_{i2} + c_{37}LZ_{i3} + d_{17}JY_i + d_{27}YB_i + d_{37}JD_i + d_{47}WD_i + e_{i7} \\ y_{i8} = a_{18} + c_{18}LZ_{i1} + c_{28}LZ_{i2} + c_{38}LZ_{i3} + d_{18}JY_i + d_{28}YB_i + d_{38}JD_i + d_{48}WD_i + e_{i8} \end{cases} \tag{6}$$

The variables in Equation (6) are:

1.  Dependent variable of forest ecological benefit

$Y_{i1}$ is annual ecological benefit of water intercepted by the forest canopy ($hm^2{\cdot}a$), $Y_{i2}$ is the water holding capacity of the forest litter ($hm^2{\cdot}a$), $Y_{i3}$ is the annual ecological benefit of water storage in the non-capillary pores of the forest soil ($hm^2{\cdot}a$), $Y_{i4}$ is the annual ecological benefit of fixing soil ($hm^2{\cdot}a$), $Y_{i5}$ is the annual ecological benefit of retaining fertilizer ($hm^2{\cdot}a$), $Y_{i6}$ is the annual ecological benefit of absorbing carbon dioxide by the forest ($m^3{\cdot}a$), $Y_{i7}$ is the annual ecological benefit of releasing oxygen by the forest ($m^3{\cdot}a$), $Y_{i8}$ is the annual ecological benefit of restraining wind and sand by the forest ($hm^2{\cdot}a$).

2.  Independent variable set of forest ecological benefit

$LF$ is a qualitative variable, whose forest type response value is: 1 for a Korean pine forest, 2 for a coniferous forest, 3 for a mixed forest, 4 for a hard broad-leaved forest, 5 for a soft broad-leaved forest and 6 for a sub arbor forest, $LZ$ is a qualitative variable, whose age group response value is: 1 for a young forest, 2 for a middle-aged forest and 3 for a mature forest, $JY$ is precipitation (mm), $YB$ is canopy closure (0, 0.1, 0.2, ... , 1), $JD$ is longitude (°), $WD$ is latitude (°), and $HB$ is altitude (m).

3.  Others are constants to be estimated

They could be estimated using observation data.

### 2.2. Monetary Construction Model of Forest Ecological Benefits

#### 2.2.1. Classification of the Monetary Quantity Construction Model

There are two alternative market technologies for forest ecological benefits:

(1) Alternative market technologies for alternative goods. According to various classical definitions of forest ecological benefits, seeking appropriate alternative goods and alternative prices, we built a monetary quantity construction model, which is called the first kind of monetary quantity construction model.
(2) Direct alternative market technology. According to the field measurement data of the external economy (or external non economy) generated by forest ecological benefits, the monetary quantity construction model of forest ecological benefits was constructed directly, which is called the second type of monetary quantity construction model.

#### 2.2.2. First Type of Monetary Construction Model

$$E_i(t) = \sum_{j=1}^{n} P_j \times R_j \times C_j \times Y(t)_{ij} \times S(t)_{ij} \dots\dots\dots\dots i = 1, 2 \dots 6; j = 1, 2 \dots n \tag{7}$$

where: $E_i$ is monetary amount of water source cultivation, fixing soil, retaining fertilizer, absorbing carbon dioxide, releasing oxygen and restraining wind and sand, $P_j$ is effective area coefficient, $R_j$ is the market approximation coefficient, $C_j$ is price of substitute goods, $Y(t)_{ij}$ is dependent variable of forest ecological benefit (physical quantity), $S(t)_{ij}$ is a forest resource vector, and $t$ is time, referring to summing according to stand and subcompartment $_j$ [36].

2.2.3. Second Type of Monetary Construction Model

The second kind of money quantity construction model is constructed from physical quantities that do not exist or for which alternative goods are difficult to find. For example, there are more than ten kinds of benefit physical quantities for a forest to improve its microclimate, but no suitable substitute commodity can be found for each. Forest noise reduction can use the reduced decibel (DB) as its physical quantity, but there is no substitute commodity. There are too many physical quantities of forest protected wildlife to establish its overall diffusion model. Forest recreation itself has direct use value and is a widened forest ecological benefit. It has no physical quantity and does not need to replace commodities [36]. The model parameters of second type monetary construction model of forest ecological benefit are shown in Table 1.

**Table 1.** Model parameters of second type money of forest ecological benefit.

| Forest Ecological Benefit | Pi | Ri | Construction Model of Money (¥/hm$^2$) * | Number of Samples |
|---|---|---|---|---|
| Improving microclimate | 0.4 | 0.8 | $E_i(t) = \sum P_j \times R_j \times S(t)_{ij} \times 67.99605 \times LZ^{0.4931957}$ | 60 |
| Reducing flood and drought | 1.0 | 0.9 | $E_i(t) = \sum P_j \times R_j \times S(t)_{ij} \times 311.6941 \times LZ^{0.6183988}$ | 30 |
| Recreation resource | 0.6 | 0.4 | $E_i(t) = \sum P_j \times R_j \times S(t)_{ij} \times 12.33866 \times LZ^{0.8235893}$ | 60 |
| Wild animal protection | 1.0 | 0.9 | $E_i(t) = \sum P_j \times R_j \times S(t)_{ij} \times 21.39681 \times LZ^{0.8760093}$ | 30 |
| Wild plant protection | 1.0 | 0.9 | $E_i(t) = \sum P_j \times R_j \times S(t)_{ij} \times 64.11374 \times LZ^{0.82359.8}$ | 60 |
| Reducing noise | 0.1 | 0.8 | $E_i(t) = \sum P_j \times R_j \times S(t)_{ij} \times 62.74023 \times LZ^{0.2500285}$ | 30 |

* Where: $E_i$ is the monetary amount of water source cultivation, fixing soil, retaining fertilizer, absorbing carbon dioxide, releasing oxygen and restraining wind and sand; $P_j$ is the effective area coefficient; $R_j$ is the market approximation coefficient; $S(t)_{ij}$ is a forest resource vector; t is time, referring to summing according to stand and subcompartment $_j$; LZ is the stand age group.

*2.3. Overall Diffusion Model*

[Definition] According to the characteristics of forest ecological benefits, it is difficult to measure them, so their measurement must be estimated by the sample. This sample cannot be random but often uses the observation value of an ecological positioning station. The sampling ratio is much smaller than that of forest resources. This extension from micro-observation data to the whole should be a gradual and layer-by-layer "diffusion". All kinds of forest ecological benefits have a correlation between multiple dependent variables and multiple independent variables. Using this multivariate-to-multivariate linear model to measure forest ecological benefits, the model that meets the overall compatibility and has independent variables was called the overall diffusion model.

The samples of the above parameter estimates are collected from the three North shelter belts, the shelter belts in the middle and upper reaches of the Yangtze River, the coastal shelter belts and the Taihang Mountains. The complexity of some factor measurements mean that many benefits have only 30 samples [30].

2.3.1. Canopy Interception

The overall diffusion model of canopy interception is:

$$I = \alpha \times EXP\left[a + \sum b_i \times LF + \sum c_i \times LZ + d\ln(JY) + f \times YB\right] \tag{8}$$

where: *I* is canopy interception (t/hm$^2$·a), *LF* is a qualitative variable, whose forest type response value is: 1 for a Korean pine forest, 2 for a coniferous forest, 3 for a mixed forest, 4 for a hard broad-leaved forest, 5 for a soft broad-leaved forest, and 6 for a sub arbor forest, *LZ* is a qualitative variable, whose age group response value: 1 for a young forest, 2 for a middle-aged forest and 3 for a mature forest, *JY* is precipitation (mm), *YB* is canopy closure (0, 0.1, 0.2, . . . , 1), $\alpha$, a, $b_i$, $c_i$, d and f are parameters to be estimated.

The estimated values of the above model parameters were: $\alpha = 6.9$, a = −0.7849, $b_1 = 0.0052$, $b_2 = −0.1834$, $b_3 = 0$, $c_1 = −0.4921$, $c_2 = −0.1919$, $c_3 = 0$, d = 0.7612, f = 1.2388, number of samples = 181.

### 2.3.2. Water Holding Capacity of the Litter

The overall diffusion model of water holding capacity of litter is:

$$K = \alpha + \beta \left[ a + \sum b_i \times LF + \sum c_i \times LZ + dJD + fWD + gHB \right] \tag{9}$$

where: $K$ is the water holding capacity of the litter (t/hm². a), $JD$ is longitude (°), $WD$ is latitude (°), $HB$ is altitude (m), $\alpha$, $\beta$, g is the parameters to be estimated. Other symbols are the same as above.

Estimated values of the above model parameters are: $\alpha = 25.374$, $\beta = 16.542$, a = 68.58, $b_1 = 4.83$, $b_2 = -1.81$, $b_3 = 0$, $c_1 = -7.42$, $c_2 = -3.04$, $c_3 = 0$, d = $-0.59$, f = 0.4415, g = 0.0015, number of samples = 181.

### 2.3.3. Soil Capillary Pore Water Storage

The overall diffusion model of capillary pore water storage in forest soil is as follows:

$$Q = a + \sum b_i \times LF + cJD + dWD + fHB \tag{10}$$

where: $Q$ is the water storage in capillary pores of the forest soil (t/hm². a). Other symbols are the same as above.

Estimated values of the above model parameters are: a = $-5085.55$, $b_1 = -254.8$, $b_2 = -72.462$, $b_3 = 0$, c = 79.80, d = $-91.8$, f = 0.75, number of samples =181.

### 2.3.4. Fixing Soil

The overall diffusion model of fixing soil is:

$$Y_1 = a + \sum b_i \times LZ + dJD + CWD + EJY + fYB \tag{11}$$

where: $Y_1$ is fixing soil (t/hm². a), for sparse forest land $Y_1 = 0$, $JY$ is precipitation (mm), $YB$ is canopy closure (0, 0.1, 0.2, . . . , 1), C, E and f are the parameters to be estimated. Other symbols are the same as above.

Estimated values of the above model parameters are: a = 4, $b_1$= 6, $b_2 = -2.567$, $b_3 = 0$, d = $-0.28445$, C = 0.87825, E = 0.01762, f = 17, number of samples = 181.

### 2.3.5. Retaining Fertilizer

The overall diffusion model of retaining fertilizer is:

$$Y_2 = a + \sum b_i \times LZ + dJD + CWD + EJY + fYB \tag{12}$$

where: $Y_2$ is the retaining fertilizer (t/hm². a), for open woodland $Y_2 = 0$. Other symbols are the same as above.

Estimated values of the above model parameters are: a = $-1.5$, $b_1 = -3.3125$, $b_2 = -0.89391$, $b_3 = 0$, d = 0.05195, C = 0.00039, E = 0.00009, f = 0.5, number of samples = 181.

### 2.3.6. Absorbing Carbon Dioxide

The overall diffusion model of absorbing carbon dioxide is:

$$Y = a + \sum b_i \times LZ + dJD + CWD + EJY + fYB \tag{13}$$

where: $Y$ is the absorbing carbon dioxide by the forest (t/m³·a). Other symbols are the same as above.

Estimated values of the above model parameters are: a = $-0.13631$, $b_1 = 0.07890$, $b_2 = 0.02197$, $b_3 = 0$, d = 0.00252, C = $-0.00293$, E = $-0.00002$, f = 0.00236, number of samples = 30.

2.3.7. Releasing Oxygen

The overall diffusion model of releasing oxygen by forest is:

$$Y = \alpha \left( a + \sum b_i \times LZ + dJD + CWD + EJY + fYB \right) \tag{14}$$

where: $Y$ is amount of oxygen released by the forest (t/m$^3$·a). Other symbols are the same as above.

Estimated values of the above model parameters are: $\alpha = 0.702$, $a = -0.13631$, $b_1 = 0.07890$, $b_2 = 0.02197$, $b_3 = 0$, $d = 0.00252$, $C = -0.00293$, $E = -0.00002$, $f = 0.00236$, number of samples = 30.

2.3.8. Restraining Wind and Sand

The overall diffusion model of forest restraining wind and sand is:

$$Y_1 = a + \sum b_i \times LZ + dJD + CWD + EJY + fYB - 1 \tag{15}$$

and

$$Y_2 = \alpha \left( a + \sum b_i \times LZ + dJD + CWD + EJY + fYB - 1 \right) \tag{16}$$

where: $Y_1$ is area of restraining wind and sand (hm$^2$/hm$^2$·a), $Y_2$ is amount of restraining wind and sand (t/hm$^2$·a). Other symbols are the same as above.

Estimated values of the above model parameters are: $\alpha = 28.7$, $a = 1.68262$, $b_1 = 0.10423$, $b_2 = 0.06526$, $b_3 = 0$, $d = 0.01376$, $C = -0.03955$, $E = -0.00067$, $f = 0.41924$, number of samples = 30.

## 3. Results

### 3.1. Estimation of the Total Physical Amount of Forest Ecological Benefits

The modeling data for this study are mainly from the database of the third forest management survey in Liangshui National Nature Reserve in 2009. The geographical coordinates are 128°47′8″–128°57′19″ E and 47°6′49″–47°16′10″ N. The annual average temperature is only −0.3 °C, the annual average maximum temperature is 7.5 °C, and the annual average minimum temperature is –6.6 °C. The accumulated temperature ≥10 °C is about 1700 °C and that ≥5 °C is about 2000 °C. The annual average precipitation is 676 mm, and the annual average number of precipitation days is 120–150 days. There are 130–150 days of snow, the annual average relative humidity is 78%, and the annual average evaporation is 805 mm. A location map of Liangshui National Nature Reserve is shown in Figure 1.

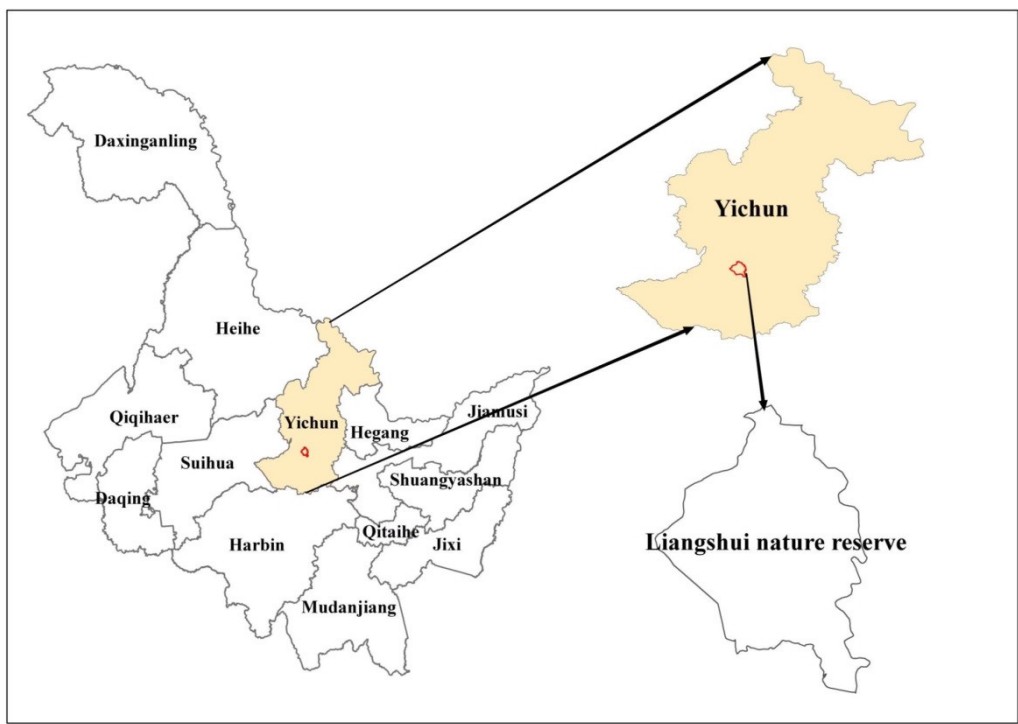

**Figure 1.** Location map of Liangshui National Nature Reserve.

Parameters of the forest ecological physical quantity model are shown in Table 2. The collection of total forest ecological benefit physics is shown in Table 3.

### 3.2. Measurement of Forest Ecological Benefits

In recent years, forest ecological benefits have expanded to the selection value and existence value of forests. For example, forest wildlife protection has value and no physical quantity, the benefit of improving the forest microclimate has an unclear physical quantity, there is no physical quantity for the benefits of forests in reducing floods and drought, and there is no substitute for forest recreation benefits and reducing noise benefits. Therefore, we directly determined the annual forest benefit per unit forest area as its value.

**Definition 1**. Under the action of atmospheric circulation, forests provide indirect forest selection value and existence value beneficial to human beings for the earth biosphere composed of life and the environment. Generally, they do not have measurable physical quantities for ecological benefits, or there are too many physical quantities to find the characteristics of "substitutes", such as forest benefits of wildlife protection and improving microclimate and other public welfare benefits. Therefore, they are called generalized forest ecological benefits.

The economic value calculated according to the generalized forest ecological benefit is shown in Table 4.

**Table 2.** Parameters of the forest ecological physical quantity model.

| Items | Canopy Interception | Litter Holding Water | Soil Holding Water | Fixing Soil | Retaining Fertilizer | Absorbing $CO_2$ | Releasing Oxygen | Restraining Wind and Sand hm$^2$ | Restraining Wind and Sand t | General Eco-Benefits |
|---|---|---|---|---|---|---|---|---|---|---|
| Para. 1 | 6.9 | 25.374 | −5085.55 | 4 | −1.5 | 0.95355 | 0.702 | 1.68262 | 28.7 | 0.4 |
| Para. 2 | −0.7849 | 16.542 | −254.8 | 6 | −3.3125 | −0.13631 | −0.13631 | 0.10423 | 1.68262 | 0.8 |
| Para. 3 | 0.0052 | 68.58 | −72.46 | −2.567 | −0.89391 | 0.0789 | 0.0789 | 0.06526 | 0.10423 | 122.6 |
| Para. 4 | −0.1834 | 4.83 | 0 | 0 | 0 | 0.02197 | 0.00293 | 0 | 0.06526 | 1 |
| Para. 5 | 0 | −1.81 | 79.8 | −0.28445 | 0.05195 | 0 | 0 | 0.01376 | 0 | 0.9 |
| Para. 6 | −0.4921 | 0 | −91.8 | 0.87825 | 0.00039 | 0.00252 | 0.00252 | −0.03955 | 0.01378 | 91 |
| Para. 7 | −0.1919 | −7.42 | 0.75 | 0.01762 | 0.00009 | −0.00293 | −0.00293 | −0.00067 | −0.03955 | 0.6 |
| Para. 8 | 0 | −3.04 | 0 | 17 | 0.5 | −0.0002 | −0.0000 | 0.41924 | −0.00067 | 0.4 |
| Para. 9 | 0.7612 | 0 | 0 | 0 | 0 | 0.00236 | 0.00236 | 0 | 0.41924 | 60.1 |
| Para. 10 | 1.2388 | −0.59 | 0 | 0 | 0 | 0 | 0 | 0 | 0 | 1 |
| Para. 11 | 0 | 0.4415 | 0 | 0 | 0 | 0 | 0 | 0 | 0 | 1 |
| Para. 12 | 0 | 0.0015 | 0 | 0 | 0 | 0 | 0 | 0 | 0 | 129 |
| Effective area coefficient | 1 | 1 | 1 | 1 | 1 | 1 | 1 | 0.5 | 0 | 0.1 |
| Market approximation coefficient | 0.8 | 0.8 | 0.8 | 0.9 | 0.1 | 1 | 0.2 | 0.8 | 0 | 1 |
| Price/ha | 0.66024 | 0.66024 | 0.66024 | 14.88 | 843.7 | 128.33 | 1269.7 | 450 | 0 | 71 |

**Table 3.** Collection of total forest ecological benefit physics.

| Age Group | Stand | Total Area hm² | Total Volume m³ | Total Physics | Water Source Cultivation y | Water Source Cultivation Total | Fixing Soil y | Fixing Soil Total | Retaining Fertilizer y | Retaining Fertilizer Total | Absorbing $CO_2$ y | Absorbing $CO_2$ Total | Releasing Oxygen y | Releasing Oxygen Total | Restraining Wind and Sand y | Restraining Wind and Sand Total |
|---|---|---|---|---|---|---|---|---|---|---|---|---|---|---|---|---|
| | | 6334 | 1,820,433 | 14,516,716 | 2215.8 | 14,034,662 | 33.2 | 210,462 | 4.7 | 29,972 | 15 | 94,913 | 8.8 | 55,794 | 0.003 | 3167.9 |
| Young | | 748 | 120,026 | 1,441,877 | 1838.5 | 1,375,208 | 40.1 | 29,961 | 2.3 | 1724 | 17.7 | 13,235 | 13 | 9743 | 0.854 | 418.4 |
| | Korean pine | 35 | 6778 | 67,724 | 1839.2 | 64,373 | 40.2 | 1407 | 2.3 | 81 | 21.4 | 748 | 15.7 | 551 | 15 | 19.7 |
| | Coniferous | 312 | 63,908 | 598,553 | 1820.2 | 567,912 | 40.4 | 12,595 | 2.3 | 722 | 22.6 | 7054 | 16.6 | 5193 | 0.567 | 176.9 |
| | Mixed | 203 | 33,853 | 412,405 | 1941.7 | 394,158 | 39.8 | 8080 | 2.3 | 466 | 18.4 | 3731 | 13.5 | 2747 | 0.941 | 112.3 |
| | Hard broad-leaved | 15 | 585 | 27,137 | 1745.5 | 26,183 | 38.7 | 581 | 2.3 | 34 | 4.3 | 64 | 3.1 | 47 | 7.467 | 8 |
| | Soft broad-leaved | 182 | 14,852 | 334,685 | 1765.1 | 321,248 | 39.9 | 7268 | 2.3 | 419 | 9 | 1638 | 6.6 | 1206 | 2.159 | 101.3 |
| | Sub arbor | 1 | 50 | 1374 | 1335 | 1335 | 29 | 29 | 2 | 2 | | | | | 14 | 0.3 |
| Middle | | 2418 | 796,924 | 5,427,877 | 2165.9 | 5,237,228 | 31.5 | 76,219 | 4.7 | 11,424 | 18.5 | 44,632 | 9.2 | 22,206 | 0.003 | 1260.2 |
| | Korean pine | 720 | 402,608 | 1,622,049 | 2155.2 | 1,551,730 | 31.3 | 22,524 | 4.7 | 3397 | 31.3 | 22,539 | 15.6 | 11,212 | 0.14 | 371 |
| | Coniferous | 999 | 276,356 | 2,240,161 | 2167.7 | 2,165,508 | 31.7 | 31,666 | 4.7 | 4725 | 15.5 | 15,487 | 7.7 | 7708 | 0.137 | 525 |
| | Mixed | 366 | 71,227 | 862,443 | 2288.9 | 837,720 | 31.5 | 11,540 | 4.7 | 1729 | 10.9 | 3991 | 5.4 | 1986 | 0.437 | 190.8 |
| | Hard broad-leaved | 26 | 3172 | 56,855 | 2124.9 | 55,247 | 31.8 | 826 | 4.7 | 123 | 6.8 | 178 | 3.4 | 88 | | 13.7 |
| | Soft broad-leaved | 307 | 43,562 | 646,369 | 2042.4 | 627,023 | 31.5 | 9664 | 4.7 | 1450 | 7.9 | 2437 | 3.9 | 1212 | | 159.7 |
| Mature | | 3168 | 903,483 | 7,646,962 | 2342.9 | 7,422,227 | 32.9 | 104,283 | 5.3 | 16,824 | 11.7 | 37,047 | 7.5 | 23,845 | | 1489.1 |
| | Korean pine | 668 | 224,243 | 1,636,089 | 2374.1 | 1,585,905 | 33.5 | 22,405 | 5.4 | 3586 | 13.6 | 9092 | 8.9 | 5914 | | 320 |
| | Coniferous | 1368 | 392,493 | 3,299,191 | 2340.8 | 3,202,150 | 33.3 | 45,524 | 5.4 | 7383 | 11.3 | 15,491 | 7.5 | 10,306 | | 638.9 |
| | Mixed | 838 | 232,454 | 2,084,974 | 2417.7 | 2,026,065 | 32.3 | 27,084 | 5.2 | 4364 | 12 | 10,017 | 7.4 | 6174 | | 392.7 |
| | Soft broad-leaved | 294 | 54,294 | 626,709 | 2068.4 | 608,108 | 31.5 | 9270 | 5.1 | 1491 | 8.3 | 2448 | 4.9 | 1450 | | 137.4 |

**Table 4.** Generalized forest ecological money benefits. Unit: 10 thousand RMB ¥.

| Age Group | Stand | Total Area hm² | Total Volume m³ | Total Money | Water Source Cultivation | Fixing Soil | Retaining Fertilizer | Absorbing $CO_2$ | Releasing Oxygen | Restraining Wind and Sand | Improving Micro Climate | Reducing Water Disaster | Recreation | Living Things Protection | Reducing Noise |
|---|---|---|---|---|---|---|---|---|---|---|---|---|---|---|---|
| | | 6334 | 1,820,433 | 9431 | 2402 | 761 | 455 | 2850 | 2550 | 103 | 45 | 93 | 16 | 147 | 8 |
| Young | | 748 | 120,026 | 1263 | 235 | 108 | 26 | 397 | 445 | 14 | 5 | 11 | 2 | 17 | 1 |
| | Korean pine | 35 | 6778 | 67 | 11 | 5 | 1 | 22 | 25 | 1 | | 1 | | 1 | |
| | Coniferous | 312 | 63,908 | 624 | 97 | 46 | 11 | 212 | 237 | 6 | 2 | 5 | 1 | 7 | |
| | Mixed | 203 | 33,853 | 355 | 67 | 29 | 7 | 112 | 126 | 4 | 1 | 3 | 1 | 5 | |
| | Hard broad-leaved | 15 | 585 | 12 | 4 | 2 | 1 | 2 | 2 | | | | | | |
| | Soft broad-leaved | 182 | 14,852 | 204 | 55 | 26 | 6 | 49 | 55 | 3 | 1 | 3 | | 4 | |
| | Sub arbor | 1 | 50 | | | | | | | | | | | | |
| Middle | | 2418 | 796,924 | 3860 | 896 | 276 | 173 | 1340 | 1015 | 41 | 17 | 36 | 6 | 56 | 3 |
| | Korean pine | 720 | 402,608 | 1635 | 266 | 81 | 52 | 677 | 512 | 12 | 5 | 11 | 2 | 17 | 1 |
| | Coniferous | 999 | 276,356 | 1440 | 371 | 114 | 72 | 465 | 352 | 17 | 7 | 15 | 3 | 23 | 1 |
| | Mixed | 366 | 71,227 | 446 | 143 | 42 | 26 | 120 | 91 | 6 | 3 | 5 | 1 | 8 | |
| | Precious hard broad-leaved | 26 | 3172 | 25 | 9 | 3 | 2 | 5 | 4 | | | | | 1 | |
| | Soft broad-leaved | 307 | 43,562 | 313 | 107 | 35 | 22 | 73 | 55 | 5 | 2 | 5 | 1 | 7 | |
| Mature | | 3168 | 903,483 | 4308 | 1270 | 377 | 255 | 1112 | 1090 | 48 | 22 | 47 | 8 | 74 | 4 |
| | Korean pine | 668 | 224,243 | 993 | 271 | 81 | 54 | 273 | 270 | 10 | 5 | 10 | 2 | 16 | 1 |
| | Coniferous | 1368 | 392,493 | 1849 | 548 | 165 | 112 | 465 | 471 | 21 | 10 | 20 | 4 | 32 | 2 |
| | Mixed | 838 | 232,454 | 1148 | 347 | 98 | 66 | 301 | 282 | 13 | 6 | 12 | 2 | 19 | 1 |
| | Soft broad-leaved | 294 | 54,294 | 319 | 104 | 34 | 23 | 74 | 66 | 4 | 2 | 4 | 1 | 7 | |

It can be seen from Table 4 that the benefits of water source cultivation, fixing soil, retaining fertilizer, absorbing carbon dioxide, releasing oxygen and restraining wind and sand of Liangshui National Nature Reserve are 24.02 million, 7.61 million, 4.55 million, 28.5 million, 25.5 million, and 1.03 million RMB ¥, respectively. The generalized ecological benefits include the benefits of improving the microclimate, reducing flood and drought, recreation, wildlife protection, and reducing noise and are valued at 930,000, 160,000, 1,470,000, and 80,000 RMB ¥, respectively, and the total annual forest ecological benefit is 94.31 million RMB ¥.

The authors also calculated the forest ecological benefits of Maoershan National Forest Park and the state-owned forest of Songhuajiang Forest Administration Bureau of Heilongjiang Province. The area of Maoershan National Forest Park is 22,720 hm$^2$, and the monetary value of forest ecological benefits is 188.27 million RMB ¥ per year. The area of the state-owned forest of Songhuajiang Forest Administration Bureau is 2,056,544 hm$^2$, and the monetary value of forest ecological benefits is 10,211.83 million RMB ¥ per year, while the area of Liangshui National Nature Reserve is only 6334 hm$^2$, and the total annual forest ecological benefits are 94.31 million RMB ¥. Calculated by unit area, the annual forest ecological benefits per hectare are Liangshui National Nature Reserve (14,889 RMB ¥) > Maoershan National Forest Park (8286 RMB ¥) > Songhuajiang Forest Administration Bureau (4965 RMB ¥). These data also show that the size of forest ecological benefits is closely related to forest quality.

## 4. Discussion

Many scholars only give the total monetary value of forest ecological benefits. We used forest management survey data to calculate and accumulate forest ecological benefits from subcompartment to subcompartment. In this way, our estimate of the amount of forest ecological benefits should be more reliable.

The concept of "overall diffusion" in the establishment of the forest ecological benefit model and the unification of various forest ecological benefits into one system was used. The first and second types of monetary models were constructed by using the market approximation coefficient, the effective area coefficient, and the physical quantity conversion coefficient, respectively, followed by estimation, which greatly reduced the controversy of overestimating the value of forest ecological benefit at home and abroad.

## 5. Conclusions

This paper proposes the concepts of classical forest ecological benefit and generalized forest ecological benefit. The forest ecological benefits with clear physical quantities such as water source cultivation, fixing soil, retaining fertilizer, absorbing carbon dioxide, releasing oxygen, and restraining wind and sand are called classic forest ecological benefits. The forest benefits without clear physical quantities or market alternative products, such as improving the microclimate, reducing flood and drought, recreation, wildlife protection, and reducing noise, are called generalized forest ecological benefits.

The six forest ecological benefits of water source cultivation, fixing soil, retaining fertilizer, absorbing carbon dioxide, releasing oxygen, and restraining wind and sand must be linear models. The theory of ecological benefit is not too deep. Therefore, it can only be described by a linear model, that is, individual exponential values and the logarithm can be transformed into a linear model. The development of these six forest ecological benefits occurred in order, thus these six forest ecological benefit models have a certain independence in that they have their own independent variable system, which is different from the multivariate linear model. These six forest ecological benefits are interrelated, so they are both independent and related, which we termed the seemingly uncorrelated linear model.

Forest management survey data were used to measure the forest ecological benefits for the first time. Firstly, the coefficient of forest ecological benefits was determined, and then the physical quantities of the forest ecological benefits were calculated by subcom-

partments. Then, the classical forest ecological benefits were obtained using the physical quantity conversion coefficient, and then the generalized forest ecological benefits were estimated using the second type of monetary model. Finally, forest ecological benefits of each subcompartment were accumulated.

Technically, based on the seemingly uncorrelated model of modern statistics, we deduced the seemingly uncorrelated model of forest ecological benefits, introduced the concept of "overall diffusion", and finally estimated the monetary value of forest ecological benefits.

**Author Contributions:** Conceptualization, H.L. and S.L.; methodology, H.L.; software, H.L.; validation, H.L. and S.L.; formal analysis, T.C.; investigation, T.C.; resources, H.L.; data curation, H.L.; writing—original draft preparation, H.L.; writing—review and editing, S.L. and T.C.; visualization, H.L.; supervision, S.L. and T.C.; project administration, S.L.; funding acquisition, T.C. All authors have read and agreed to the published version of the manuscript.

**Funding:** This research was funded by the Scientific Research Project of Central University, grant number 2572020DR11, and the General Project of National Natural Science Foundation of China, grant number 31970385. The APC was funded by the Young and Middle-aged Teacher Development Program of Shanghai Vocational College of Agriculture and Forestry, grant number A2-0265-22-28.

**Institutional Review Board Statement:** Not applicable.

**Informed Consent Statement:** Not applicable.

**Data Availability Statement:** Data was obtained from the stakeholders of this project is available upon request to the authors with the permission of the third parties.

**Acknowledgments:** The authors are grateful for the technical support of Changsheng Li, working at the Harbin Institute of Technology, People's Republic of China.

**Conflicts of Interest:** The authors declare no conflict of interest. The funders had no role in the design of the study; in the collection, analyses, or interpretation of data; in the writing of the manuscript, or in the decision to publish the results.

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
