# Peer review of "Measurement of Forest Ecological Benefits Based on Big Data"

_sustainability, doi:10.3390/su14127248_

Round 1

Reviewer 1 Report

This paper proposes an effective modern forest eco-efficiency estimation model to estimate the forest eco-efficiency of the Liangshui National Nature Reserve using the forest big data from the Third Forest Management Survey.  I believe that it is an interesting paper on this topic. However, there are a few concerns before I recommend it for publication.

1) The paper lacks a general overview of the study area. It is suggested that a map of the geographical location of the study area be included before the introduction of the methodology section, and a brief discussion of the ecological elements of the study area and the significance of the study, etc.

2) The introduction part of the research process and methods in this paper is not clear, and it lacks an overall flow chart describing the general methods and several paragraphs outlining the general methods, rather than just listing the detailed equations.

3) The structure is not well organized and the general readability is poor. Such as Tables 3 and 4 both mentioned the ecological benefits of different physical quantities at different “Age group”,  but I do not see any analysis of “Age group” which make the whole paper seems inconsistent.

4) Formatting of the formulas is poor, the authors should check all the equations regarding the position, font size, italics, serial numbers, etc to meet the requirements of the journal.

5) Referencing does not seem like the style of sustainability, please read the check the whole section.

6) line 274, this should be part of the main text.

Reviewer 2 Report

The article deals with the issue of monetary value of ecosystem benefits of forest stands. The text is written clearly and intelligibly. In terms of content, however, it has the following major shortcomings:

The abstract contains a lot of well-known information. I recommend significantly shortening the first half of the abstract. Information on the methodology used is practically missing here.

However, I see the main problem in using the term "The service function of forest ecosystem". However, current scientific knowledge distinguishes between the terms "ecosystem function" and "ecosystem service". The authors should unify the terminology (I recommend studying more literature).

Line 143 "Others are constants to be estimated" - Oki, but estimated based on what?

Table 1 - It is not clear where the authors took the numbers and powers in the samples (eg 67.99605 or 0.4931957)

Subchapter 2.3. - It is good to explain the estimate in more detail - on the basis of which it was done

The methodology is elaborated in detail. However, a closer presentation of the input data is missing. Without them, the results are inconclusive.

The Conclusions chapter lacks more fundamental findings (only superficial findings are given)

Chapter conclusions  - „Many scholars only give a total monetary vaule od forest ecological benefits“ –  However, there are also approaches that evaluate individual ecosystem services, such as VYSKOT, Ilja, et al. Quantification and evaluation of forest functions on the example of the Czech Republic. Ministry of Environment of the Czech Republic, 2004.

Overall, recommend studying more literary sources from more countries (in the current overview, Chinese sources dominate, sources from other countries are minimal).

Round 2

Reviewer 1 Report

The authors have improved the paper significantly. However, the general readability and some formatting/spelling are still there. Also, the conclusion part is ahead of the discussion part which is not correct.

Author Response

Dear Reviewer

We really appreciate your comments and suggestions. We have re-organized the language and revised the manuscript following the suggestions The manuscript was revised by a professional language editing service (Elixigen Co.). We hope this version would be much better and be satisfied. Furthermore, we exchanged the conclusion part and the discussion part.

Once again, thank you very much!

Reviewer 2 Report

The authors modified the article according to the comments and incorporated the required changes. However, it would still be appropriate to expand the literature review. This list and overview of the issue lacks more work with a similar approach to forest ecosystem services.

Author Response

Dear Reviewer

We really appreciate your comments and suggestions. We have studied more literary sources, especially the study of the valuation methods. We added some information about the similar approach to forest ecosystem services as suggested. We also have re-organized the language and further revised the manuscript. The language was extensively edited by the Elixigen Co.. We hope you are satisfied with the new version.

Once again, thank you.
